# FuzzyCLIP: Clustering-Driven Stacked Prompt in Zero-Shot Anomaly Detection

## Abstract

How to enhance the alignment of text and image features in CLIP model is a key challenge in zero-shot industrial anomaly detection tasks. Recent studies mostly rely on precise category prompts for pre-training, but this approach is prone to overfitting, which limits the generalization ability of the mode. To address this issue, we propose the concept of fuzzy prompts and introduce Clustering-Driven Stacked Prompts (CSP) along with the Ensemble Feature Alignment (EFA) module to improve the alignment between text and image features. This design significantly outperforms other methods in terms of training speed, stability, and final convergence results, showing remarkable efficiency in enhancing anomaly detection segmentation performance. What is even more surprising is that fuzzy stacked prompts exhibit strong generalization in classification tasks, enabling them to adapt to various anomaly classification tasks without any additional operations. Therefore, we further propose the Regulating Prompt Learning (RPL) module, which leverages the strong generalization ability of fuzzy stacked prompts to regularize prompt learning, thereby improving performance in anomaly detection classification tasks. We conducted extensive experiments on seven industrial anomaly detection datasets, which demonstrate that our method achieves state-of-the-art performance in zero-shot anomaly detection and segmentation tasks.

Industrial anomaly detection (Xie et al., 2023; Roth et al., 2022; Mou et al., 2023; Wang et al., 2023) is an important research area in the field of computer vision. It involves two primary tasks: distinguishing between normal and anomalous images through anomaly detection classification and achieving pixel-level anomaly localization through anomaly detection segmentation. However, the challenge of anomaly detection lies in the unknown nature of anomalies. Due to the lack of available anomalous data samples, it is difficult to extract features of anomalies. Traditional methods often rely on unsupervised (Yi & Yoon, 2020; Massoli et al., 2022; Sohn et al., 2021; Gong et al., 2019) or self-supervised (Deng & Li, 2022a; Zhu et al., 2024; Deng & Li, 2022b; Cao et al., 2022) approaches. By learning from a large number of normal samples, the model can memorize the characteristics of these normal samples and then detect anomalies by calculating the differences between test samples and the learned normal distribution. A significant drawback of these methods is the necessity of handling large amounts of cross-class data. Furthermore, as the number of object categories to be detected increases, a separate model needs to be trained for each category, leading to a significant increase in the number of models. Therefore, developing a unified cold-start model that can adapt to multiple categories without requiring additional training becomes an ideal solution and is also an open challenge faced by both the academic and industrial communities. Recently, significant progress has been made in zero-shot anomaly detection (ZSAD) (Baugh et al., 2023; Deng et al., 2023; Cao et al., 2023) using CLIP (Radford et al., 2021) as a pre-trained vision-language model (Cao et al., 2024; Zhou et al., 2024; Chen et al., 2023; Jeong et al., 2023; Chen & et al., 2023). This approach identifies anomalies by computing the cosine similarity between text and image features, marking regions with higher similarity as anomalous. A pioneering ZSAD work, WinCLIP (Jeong et al., 2023), employed manually designed text templates and multi-scale image feature extraction for classification tasks, achieving excellent results by aligning image and text features. Subsequently, APRIL-GAN (Chen & et al., 2023) introduced a strategy enhanced by a pre-trained linear layer, utilizing features from different layers to align text and image features, demonstrating outstanding performance in anomaly segmentation tasks as well. Following APRIL-GAN (Chen & et al., 2023), several methods based on pre-trained linear layers or adapters have emerged for segmentation tasks. On the template of text prompts, WinCLIP (Jeong et al., 2023)

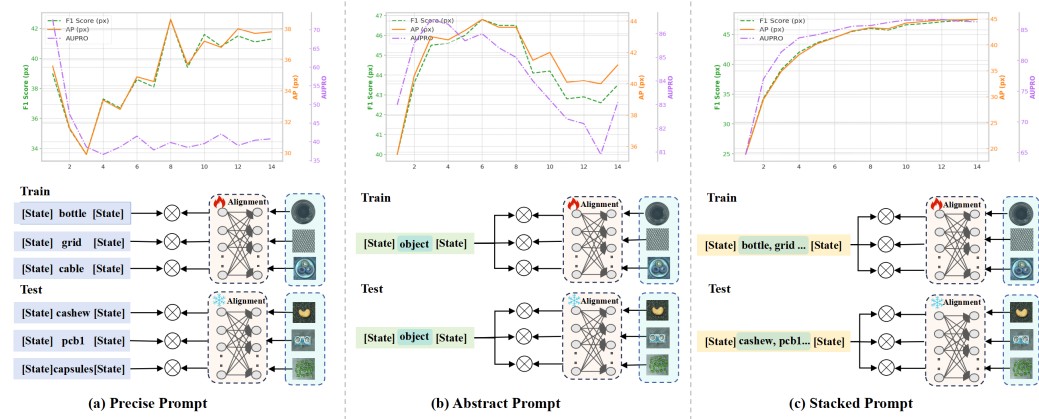

Figure 1: We compared several different prompt design methods. (a) uses precise prompts, leading to significant fluctuations in various metrics during training and severe overfitting, which results in a notable decline in the AUPRO metric. (b) adopts a uniform abstract fuzzy prompt, replacing specific class names with "object". During training, this method quickly achieves fitting but also experiences overfitting; however, its convergence outcome is much better than that of the precise prompts. (c) illustrates our proposed stacked fuzzy prompt method. As observed, this method surpasses the previous methods in terms of stability, convergence speed, and final convergence results, demonstrating a remarkably strong generalization capability.

used a completely training-free approach with precisely designed text templates for different categories, obtaining normal and anomalous text features by averaging. APRIL-GAN (Chen & et al., 2023) and subsequent methods (Zhou et al., 2024; Chen et al., 2023; Cao et al., 2024) adopted this design, using precise category prompts during both training and testing phases. However, as shown in Fig 1, the use of precise category prompts led to poor stability in the model during training, with significant fluctuations in various metrics. We believe this instability is due to the linear layer learning various object categories' related semantics during training. To address this issue, we experimented with fuzzy text prompts, replacing all categories with the abstract noun "object". The results showed that the model converged rapidly within less than one epoch, with metrics surpassing those of the original APRIL-GAN (Chen & et al., 2023) results, though overfitting occurred subsequently. Based on these tests, we propose a Clustering-Driven Stacked Fuzzy Prompt approach. By stacking object categories and using fuzzy text prompts, this method effectively avoids overfitting caused by abstract prompts and trains multiple linear layers capable of learning different knowledge in Fig 1 demonstrates the performance of the Clustering-Driven Fuzzy Stacked Prompt in anomaly detection segmentation tasks. Compared to the method of using precise category prompts, it is possible to achieve outstanding stability, convergence speed, and final convergence results. After clustering training, different linear layers learn different abnormal feature attributes. During the testing phase, different weights are assigned to the different linear layers based on the distance between the test class and the training cluster centers, thus more comprehensively segmenting out abnormal areas.

In the anomaly classification task, AnomalyCLIP (Zhou et al., 2024) achieved notable results using prompt learning (Zhou et al., 2021; 2022; Khattak et al., 2023b; Li et al., 2024). By pre-training a class-agnostic prompt, it not only eliminates the need for manually designed templates but also avoids the influence of object semantics on the alignment process, thereby better aligning the anomalies themselves. However, a major issue with prompt learning in classification tasks is its susceptibility to overfitting. To prevent overfitting, AnomalyCLIP (Zhou et al., 2024) had to reduce the number of training parameters, which in turn limited its performance. In our tests, we found that fuzzy stacked prompts showed certain advantages in classification tasks, even without any training, by directly aligning with image features. It is important to note that precise prompts generate two text features for each category, whereas fuzzy prompts generate only two text features for all categories, one positive and one negative. Therefore, to avoid overfitting in prompt learning for classification tasks and to fully leverage the advantages of prompt learning, we employed fuzzy

stacked prompts for regularization. This approach not only improves the stability and generalization of the model but also enhances its performance in classification tasks.

Finally, we conducted extensive experiments to verify the effectiveness of our mixed prompts in adapting the base model to zero-shot anomaly segmentation. Specifically, our final model, Fuzzy-CLIP, achieved state-of-the-art performance on various zero-shot anomaly segmentation datasets under different settings. Our contributions are summarized as follows:

- We propose the concept of fuzzy prompts based on the CLIP model and apply it to industrial anomaly detection tasks.

- We proposed Clustering-Driven Stacked Prompt (CSP) and Ensemble Feature Alignment (EFA) modules using fuzzy stacked prompts for pre-training, enhancing feature alignment, and achieving accurate anomaly segmentation.

- We introduced Regulating Prompt Learning (RPL), using fuzzy stacked prompts to regularize prompt learning and complete anomaly classification.

- The comprehensive experimental results on multiple datasets in the industrial anomaly detection field indicate that FuzzyCLIP has achieved excellent zero-shot anomaly detection performance in handling highly diverse semantic data from the defect detection domain.

# 1 RELATED WORKS

**Zero-shot Anomaly Detection.** In the field of industrial anomaly detection, pre-trained vision models (Dosovitskiy et al., 2021; Li et al., 2022; Radford et al., 2021; Khattak et al., 2023a; Zhang et al., 2023) have demonstrated strong performance due to their excellent generalization and feature extraction capabilities. Currently, anomaly detection methods based on pre-trained large models can be categorized into two main types: The first type does not require any additional training, such as WinCLIP (Jeong et al., 2023) and SAA (Cao et al., 2023). As a pioneering work, WinCLIP (Jeong et al., 2023) employs a sliding window method to extract multi-granularity image features for feature alignment, achieving significant results in classification tasks. However, it requires multiple encodings of the same image to capture anomalous features. SAA (Cao et al., 2023), as a pioneer in the collaboration of multiple pre-trained large models, combines the capabilities of Grounding DINO (Liu et al., 2023) and SAM (Khattak et al., 2023a), where Grounding DINO achieves localization through text prompts, followed by SAM performing segmentation using box prompts. However, a notable drawback of this method is its high usage cost and long inference time.

The second type of method requires additional training on anomaly detection data. APRIL-GAN (Chen & et al., 2023) first proposed using a linear layer to enhance the alignment of text features with image features at different levels, successfully completing anomaly segmentation tasks, but it overlooked the classification task. Similarly, SDP (Chen et al., 2023) uses a linear layer to strengthen feature alignment and incorporates CLIP Surgery (Li et al., 2023) with a V-Vattention dual-branch structure. Although this approach significantly improves anomaly perception, the dual-branch structure introduces additional computational costs.

**Prompt Learning in Vision-Language Models.** Prompt Learning, as an efficient alternative to parameter tuning, differs from traditional full-network fine-tuning by achieving satisfactory results with fewer tuned parameters. CoOp (Zhou et al., 2021) introduced learnable text prompts for few-shot classification. Building on this, DenseCLIP (Rao et al., 2022) extended prompt learning to dense prediction tasks by adding an image decoder. PromptSRC (Khattak et al., 2023b) introduced regularization through raw feature output, while AnomalyCLIP (Zhou et al., 2024) became the first model to apply prompt learning to industrial anomaly detection, proposing object-agnostic prompt learning to avoid the potential adverse effects of different object semantics on anomaly detection. With its glocal context optimization, AnomalyCLIP(Zhou et al., 2024) is capable of capturing local anomalous semantics, thus allowing it to perform both classification and segmentation tasks without the need for an additional decoder network. However, the dual-branch structure of AnomalyCLIP(Zhou et al., 2024) is undoubtedly its greatest drawback, increasing model complexity and computational costs. Moreover, the pre-training approach may lead to underfitting or overfitting issues, posing challenges to the model's stability and generalization capabilities.

## 2 APPROACH

### 2.1 OVERVIEW

In this paper, we propose FuzzyCLIP, which enhances segmentation and classification performance in industrial anomaly detection through stacked fuzzy prompts. As shown in Fig 2, FuzzyCLIP first introduces the Clustering-Driven Stacked Prompt (CSP) module to categorize the training data (see Sec.2.2). It then employs the Ensemble Feature Alignment (EFA) module (see.2.3) to learn different anomalous features, further improving the alignment of image and text features. For anomaly detection classification tasks, we design the Regulating Prompt Learning (RPL) module, which utilizes the broad generalization ability of stacked fuzzy prompts to regularize prompt learning (see Sec.2.4), thereby effectively enhancing classification performance.

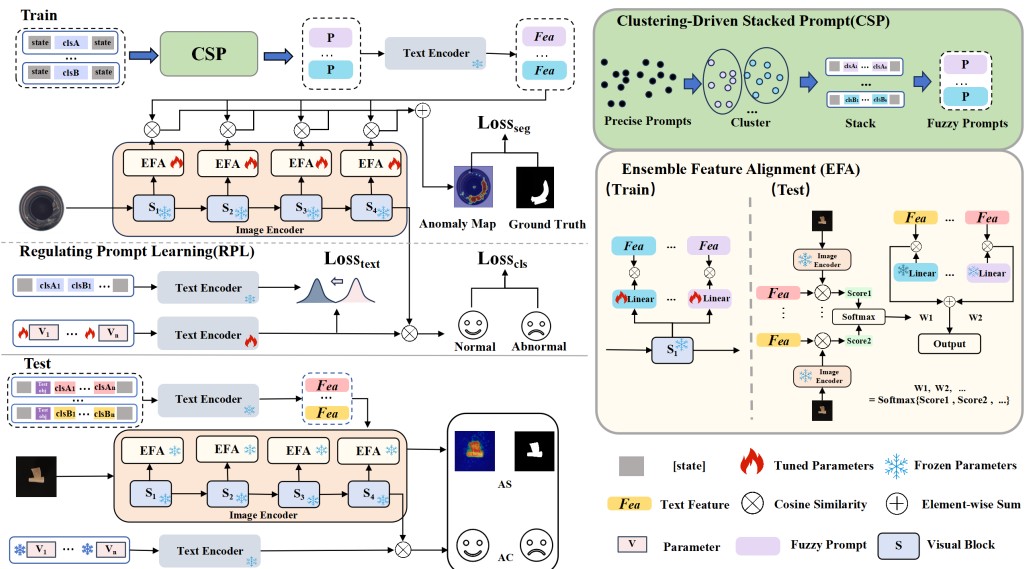

Figure 2: Overview of FuzzyCLIP. To enhance the alignment of image and text features and accomplish anomaly detection and segmentation tasks, FuzzyCLIP introduces Clustering-Driven Stacked Prompts (CSP) and an Ensemble Feature Alignment (EFA) module to learn various anomalous attributes. In anomaly detection classification tasks, we regularize prompt learning through stacked fuzzy prompts, a process referred to as the Regulating Prompt Learning (RPL) module, which effectively improves classification performance and enhances the model's generalization ability.

### 2.2 CLUSTERING-DRIVEN STACKED PROMPT (CSP)

In the application of CLIP, text descriptions play a critical role. One of the most widely used techniques is the Compositional Prompt Ensemble (CPE), introduced by the pioneering work of Win-CLIP (Jeong et al., 2023). CPE generates different descriptions by designing positive and negative templates. Specifically, CPE combines predefined states and template lists, inserting object names into these templates.

$$\text{Prompt} = [State] \, [\boldsymbol{Cls}] \, [State] \tag{1}$$

After passing through the text encoder of CLIP (Radford et al., 2021), the positive and negative text features are averaged, resulting in two sets of text features, $\boldsymbol{T}_n$ and $\boldsymbol{T}_p$. Additionally, the cosine similarity between the two representative vectors and the image feature $\boldsymbol{F}_c$ is used to determine which distribution the image is more inclined towards, indicating whether the object is more likely to be normal or anomalous.

$$S = \text{Softmax}\left(\boldsymbol{F}_c \cdot [\boldsymbol{T}_n, \boldsymbol{T}_p]^T\right) \tag{2}$$

The goal of CPE design is to achieve feature alignment without the need for training. Later methods that involve training, such as APRIL-GAN (Chen & et al., 2023), inherited the CPE design concept

but still used class-specific CPE during training. This led to severe fluctuations during training and poor generalization of the trained feature alignment enhancement modules (such as linear layers). These methods also used $n * 2$ sets of text features. Based on these issues, we propose class-stacked fuzzy prompts to improve the stability and generalization of feature alignment.

$$\text{StackedPrompt} = [State] [\boldsymbol{Cls_a}] \ldots [\boldsymbol{Cls_n}] [State] \tag{3}$$

In our improvement, the $[State]$ part still employs the CPE method to design templates and process text features using averaging to obtain two text features, positive and negative.

For class stacking, we use K-means clustering on all categories in the training data and design a scoring mechanism to select the stacked classes. In our improvement, we use the total sum of squared distances from each class to its cluster center, plus a penalty factor, to calculate the score. We then select the number of classes with the lowest score.

$$\boldsymbol{k^*} = \arg\min_k \left( \sum_{i=1}^{k} \sum_{x \in C_i} \|x - \frac{1}{|C_i|} \sum_{x' \in C_i} x'\|^2 + \boldsymbol{\lambda(n)} \right) \tag{4}$$

In this context, $|C_i|$ represents the number of data points in class $C_i$. Additionally, we introduced a penalty coefficient $\lambda(n) = 0.1 \times e^n$ to optimize the model's performance during the clustering and feature alignment process. The introduction of this penalty factor $\lambda(n)$ is intended to balance the number of classes, as each additional class requires training an extra set of linear layers in subsequent modules. Having too many classes not only reduces the amount of data per class, affecting training effectiveness but also increases computational and training complexity. By using this scoring mechanism, we optimize the number of classes and enhance the effectiveness of feature alignment.

## 2.3 ENSEMBLE FEATURE ALIGNMENT (EFA)

For anomaly detection segmentation, extracting features from different layers of an image encoder to obtain diverse image features is an efficient approach. However, while CLIP links image features and text features in a joint embedding space, during training, only class labels receive direct supervision from language signals, whereas the entire image feature map lacks similar guidance. In other words, the alignment between image feature maps and text features is missing, making direct comparison to infer anomaly maps unfeasible.

To enhance the alignment between text features and image features, we adopted pre-trained linear layers similar to APRIL-GAN. However, we obtained multiple groupings and their corresponding fuzzy stacked text prompt features through the CSP module. Based on these groupings, we trained multiple linear layers separately.

$$\boldsymbol{F_c^{j'}} = k^j \boldsymbol{F_c^j} + b^j \tag{5}$$

In this context, $F_s^{j'} \in \mathbb{R}^{H \times W \times C}$ represents the image features output from different layers $j$, which we typically obtain from the layers at indices [6, 12, 18, 24]. $k^j$ and $b^j$ represent the weights and biases of the linear layer at level $j$, respectively. We then compute the cosine similarity between the textual features and image features, and after applying softmax normalization, we obtain the anomaly map results $M_f^j$ for each layer. This prepares us for the subsequent training.

$$M_f = \text{Softmax} \left( \boldsymbol{F_c^{j'}} \cdot [\boldsymbol{T_n}, \boldsymbol{T_p}]^T \right) \tag{6}$$

During training, we froze the parameters of CLIP and used focal loss ($\mathcal{L}_{\text{focal}}$) and dice loss ($\mathcal{L}_{\text{dice}}$) functions to optimize the linear layers. This approach aims to improve the alignment between text features and image features, thereby enhancing the performance of anomaly detection segmentation.

$$\mathcal{L}_{\text{focal}} = - \alpha(1 - M_f)^\gamma \log(M_f) M_{\text{gt}}$$
$$- (1 - \alpha)M_f^\gamma \log(1 - M_f)(1 - M_{\text{gt}}) \tag{7}$$

$$\mathcal{L}_{\text{dice}} = 1 - \frac{2 \sum(M_f \cdot M_{\text{gt}}) + \epsilon}{\sum(M_f) + \sum(M_{\text{gt}}) + \epsilon} \tag{8}$$

where $M_{\text{gt}}$ is the ground truth anomaly map and the hyperparameters $\alpha$, $\gamma$, and $\epsilon$ are set to 1, 2, and 1, respectively. The final loss function is $\mathcal{L} = \mathcal{L}_{\text{focal}} + \mathcal{L}_{\text{dice}}$.

During the testing phase, we employ different fuzzy stacked prompts for inference. By generating multiple sets of textual prompt features based on the stacking method between test and training categories, we adapt to multiple linear layers.

$$\text{StackedPrompt}_i = [State]\,[\boldsymbol{Cls_{test}}]\,[\boldsymbol{Cluster_i}]\,[State] \tag{9}$$

In the formula, $Cluster_i$ represents the category names from the clustering performed during training. Since the features captured by different linear layers vary, for test samples of different categories, we assign weights to the outputs of each linear layer by calculating the cosine similarity between the image features and the clustered textual prompt features generated during training. Next, we compute the cosine similarity between the weighted outputs of the textual and image features, and by summing the multiple outputs, we obtain the final anomaly detection map. This approach fully leverages the advantages of each linear layer, allowing the model to handle anomaly detection and segmentation tasks more flexibly and accurately across different categories.

$$T_n^i = \text{TextEncoder}\{\text{StackedPrompt}_i\} \tag{10}$$

$$w_i = \text{softmax}(F_c \cdot T_n^i),\,\text{Output} = \sum_{i=1}^{k} w_i \cdot \text{Output}_i \tag{11}$$

where $T_n^i$ represents the text feature obtained from the stacked prompt through the text encoder, $F_c$ denotes the image features. $w_i$ represents the weight assigned to each linear layer. This approach effectively leverages the strengths of each linear layer, allowing the model to handle anomaly detection segmentation tasks more flexibly and accurately when dealing with different categories.

## 2.4 Regulating Prompt Learning (RPL)

In anomaly detection classification tasks, manually designed text templates often struggle to accurately generate text embeddings that capture both anomalous and normal semantics, thereby affecting the effective querying of corresponding visual embeddings. Furthermore, the text features $T_s \in \mathbb{R}^{n \times 2 \times C}$ generated by the Compositional Prompt Ensemble (CPE) increase computational complexity. However, our tests reveal that fuzzy stacked text prompts exhibit strong capabilities. Notably, the text features $T_s' \in \mathbb{R}^{1 \times 2 \times C}$ generated by the fuzzy stacked text prompts after passing through the text encoder can be applied to multiple categories, as shown in Table 1. To ad-

Table 1: Classification Performance Metrics on MVTec Dataset

| No Train | Classification (MVTec) | | | Prompt |
| | AUROC | AP | $F_1$-max | |
| --- | --- | --- | --- | --- |
| Precise Prompt | 86.1 | 93.5 | 90.4 | $\mathbb{R}^{n \times 2 \times C}$ |
| Fuzzy Prompt | **87.7** | **94.6** | **90.9** | $\mathbb{R}^{1 \times 2 \times C}$ |

dress this issue, we leveraged the strong generalization capability of fuzzy stacked prompts. In prompt learning, in addition to introducing a loss function for the classification task, we also added a regularization loss that uses the text features from the fuzzy stacked prompts to regularize the prompt. Specifically, in the anomaly detection classification task, cross-entropy loss is employed to enhance the generalization performance of the classification task. To further optimize the model, mean squared error loss is introduced to regularize the prompts through fuzzy stacked prompts. This strategy combines the classification capability of cross-entropy loss with the regularization effect of mean squared error loss, thereby improving the model's generalization ability, reducing the risk of overfitting, and enhancing classification accuracy.

$$\mathcal{L}_{\text{cls}} = -\log(p_y), \quad \mathcal{L}_{\text{text}} = \frac{1}{d}\sum_{i=1}^{d}(T' - T_{\text{train}})^2 \tag{12}$$

where $p_y$ is the predicted probability for the true label $y$, $d$ is the feature vector dimension, $T'$ the fuzzy stacked prompt feature vector, and $T_{\text{train},i}$ the trained text feature vector. During the testing phase, this single set of trained prompts is sufficient to perform anomaly classification for all classes.

## 3 EXPERIMENTS

### 3.1 EXPERIMENTAL SETUP

We conducted a series of experiments to assess the anomaly segmentation performance of our method in a zero-shot setting, focusing on the latest and most challenging industrial anomaly segmentation benchmarks. We also performed extensive ablation studies to validate the effectiveness of each component we proposed.

#### 3.1.1 DATASETS AND METRICS.

We conduct experiments on seven real industrial datasets, including MVTec-AD (Bergmann et al., 2019), VisA (Zou et al., 2022), BTAD (Mishra et al., 2021) , MPDD (Jezek et al., 2021), DAGM (Wieler & Hahn, 2007), KSDD (Tabernik et al., 2020) and DTD-Synthetic (Aota et al., 2023). We conducted a fair and comprehensive comparison with existing zero-shot anomaly detection and segmentation (ZSAS) methods using widely adopted metrics, namely AUROC, AP, AUPRO, and $F_1$-max. The anomaly detection performance is evaluated using the Area Under the Receiver Operating Characteristic Curve (AUROC). AP quantifies the accuracy of the model at different recall levels. The PRO metric represents the coverage of the segmented region over the anomalous region. $F_1$-max represents the harmonic mean of precision and recall at the optimal threshold, implying the model's accuracy and coverage.

#### 3.1.2 IMPLEMENTATION DETAILS.

We use the publicly available CLIP model (VIT-L/14@336px) as our backbone and extract patch embeddings from 6-th, 12-th, 18-th, and 24-th layers. The images used for training and testing are scaled to a resolution of $518 \times 518$. The length of learnable word embeddings is set to 12. The learnable token embedding is attached to the first 8 layers of the text encoder, with a length of 29 in each layer. All parameters of the CLIP model are frozen. Due to the ZSAD task, it is necessary to ensure that the auxiliary data does not contain the content of the test dataset. The framework training employ the Adam optimizer. For the linear training sets MVTec-AD and VisA, the learning rates are set at 1e-4 and 1e-3 for the this stage, respectively, while in the prompt learning stage, both are set at 1e-4. Training proceeds for 2 epochs with a batch size of 16. In the RPL phase, we set the length of the learnable word embeddings to 12. These learnable token embeddings are appended to the first 9 layers of the text encoder to refine the text space, with each layer having a length of 20. The entire training process lasts for one epoch and the learning rate is set at 1e-3. All experiments were conducted using PyTorch 1.10.0 and run on a single NVIDIA RTX 3090 24GB GPU.

### 3.2 PERFORMANCE COMPARISON WITH SOTA METHOD

We compared methods without the need for training, such as WinCLIP (Jeong et al., 2023), SAA (Cao et al., 2023), and CLIP Surgery (Li et al., 2023), to those that necessitate training, including APIRL-GAN (Chen & et al., 2023), CLIP-AD (Chen et al., 2023), AnomalyCLIP (Zhou et al., 2024). We use the experimental results from the original paper, and since the CLIP Surgery and AnomalyCLIP methods only have some metrics, we reproduce the results using the original code and the weight files provided in the code. As shown in Table 2, our method outperforms other approaches on all metrics for segmentation tasks in both the MVTec and VisA datasets under the zero-shot configuration. In classification tasks, our method slightly lags behind SDP+ in terms of AUROC and $F_1$-max metrics. This is primarily due to SDP+ employing a dual-branch structure, which enhances performance through multi-branch collaboration. Nevertheless, our method exceeds or matches all metrics of AnomalyCLIP, which also uses a dual-branch structure, but employs prompt learning. Notably, while our method is slightly below SDP+ on the MVTec dataset, it significantly surpasses SDP+ on all metrics for the VisA dataset. To further validate the effectiveness of our method, we tested it on other public datasetsBTAD (Mishra et al., 2021) , MPDD (Jezek et al., 2021), DAGM (Wieler & Hahn, 2007), KSDD (Tabernik et al., 2020) and DTD-Synthetic (Aota et al., 2023) with results shown in Table 3. All methods were trained on the VisA dataset. This allows for the comparison of performance across different datasets and provides a more comprehensive evaluation of the effectiveness of FuzzyCLIP. Our method demonstrates strong capabilities in both anomaly detection segmentation and classification tasks. While AnomalyCLIP, which employs

a class-agnostic training approach, also achieves relatively good results, it does not perform as well as FuzzyCLIP in terms of overall segmentation effectiveness and classification accuracy (AP).

Table 2: Performance comparison of SOTA approaches on the MVTec-AD (Bergmann et al., 2019) and VisA (Zou et al., 2022) datasets. Evaluation metrics include AUROC, $F_1$-max, AUPRO, and AP. Bold indicates the best performance and underline indicates the runner-up.

| Evaluation Type | Method | WinCLIP | APRIL-GAN | CLIP Surgery | SAA+ | SDP+ | AnomalyCLIP | FuzzyCLIP (Ours) |
|---|---|---|---|---|---|---|---|---|
| **Pixel-Level** | MVTec | (64.6,18.2,31.7) | (44.0,40.8,43.3) | (69.9,23.2,29.8) | (42.8,37.8,28.8) | (85.1,36.3,40.0) | (81.4,34.5,39.1) | (**86.4,46.0,47.6**) |
| (AUPRO,AP,$F_1$-max) | VisA | (56.8, 5.4 ,14.8) | (86.8,25.7,32.3) | (64.7,10.3,15.2) | (36.8,22.4,27.1) | (83.0,18.1,24.6) | (87.0,21.3,28.3) | (**89.8,28.0,34.2**) |
| **Image-Level** | MVTec | (91.8,96.5,92.9) | (86.1,93.5,90.4) | (90.2,95.5,91.3) | (63.1,81.4,87.0) | (**92.2,96.6,93.4**) | (91.5,96.6,92.7) | (91.7,96.6,92.7) |
| (AUROC,AP,$F_1$-max) | VisA | (78.1,81.2,79.0) | (78.0,81.4,78.7) | (76.8,80.2,78.5) | (71.1,77.3,76.2) | (78.3,82.0,79.0) | (82.1,85.4,80.4) | (**84.7,86.9,82.7**) |

In Fig 3, we present visual results of zero-shot anomaly segmentation (ZSAS) to further validate the effectiveness of our proposed method. We also compare our approach with other methods such as SAA+, APRIL-GAN, SDP+, and Anomaly-CLIP. In comparison to these methods, our approach demonstrates stronger performance in both localization and segmentation of anomaly regions, providing more accurate identification of anomalous areas and yielding superior segmentation results.

Table 3: Performance Comparison of Different Methods across Various Tasks

| Task | Method | BTAD | DAGM | DTD | SDD | MPDD | Average Rank |
|---|---|---|---|---|---|---|---|
| **Pixel-level** | APRIL-GAN | (21.9,32.4,37.4) | (21.8,47.5,50.3) | (41.5,67.7,65.4) | (17.5,15.0,25.9) | (27.8,24.9,29.7) | 2.9 |
| (AUPRO,AP,$F_1$-max) | AnomalyCLIP | (66.0,43.2,49.4) | (**88.6,58.1,59.6**) | (87.9,52.8,55.9) | (**91.0,41.7,50.0**) | (80.2,27.8,32.7) | 1.8 |
| | Ours | (**73.0,45.9,49.9**) | (79.1,47.6,51.8) | (**91.6,68.5,67.0**) | (87.2,23.6,36.6) | (**88.6,29.2,33.1**) | **1.4** |
| **Image-level** | APRIL-GAN | (69.7,21.9) | (94.5,95.8) | (94.0,85.5) | (88.0,96.7) | (**82.5,76.8**) | 2.4 |
| (AUROC,AP) | AnomalyCLIP | (**85.2,87.9**) | (95.8,97.8) | (94.6,93.9) | (80.0,95.8) | (80.4,75.8) | 1.8 |
| | Ours | (83.2,83.5) | (**96.3,96.7**) | (**96.5,91.6**) | (**93.0,97.3**) | (76.8,76.2) | **1.7** |

## 3.3 ABLATION STUDIES

To validate the effectiveness of our method, we conducted a component-wise analysis of the prompt design in our framework. All ablation studies are conducted on the Visa.

### 3.3.1 THE EFFECTIVENESS OF FUZZY PROMPT

We tested the fuzzy prompts under the same settings as APRIL-GAN, making only changes to the text prompts without any other modifications. This approach allowed us to evaluate the performance of fuzzy prompts under identical conditions, ensuring their effectiveness and improvement in the specific task.

Table 4: Comparison between precise prompts, abstract fuzzy prompts, and stacked fuzzy prompts.

| Method | Segmentation | | | epoch |
|---|---|---|---|---|
| | **AUPRO** | **AP** | **$F_1$-max** | |
| Precise Prompt | 44.0 | 40.8 | 43.3 | 15 |
| Abstract Fuzzy Prompt | 83.2 | 42.0 | 44.2 | 2 |
| Stacked Fuzzy Prompt | **86.6** | **44.2** | **46.6** | **2** |

The results in Table 4 clearly demonstrate that whether using abstract fuzzy prompts (i.e., replacing specific categories with abstract nouns) or stacked fuzzy prompts, the fuzzy prompts significantly outperform precise text prompts in terms of both convergence speed and final convergence results. This indicates that fuzzy prompts not only accelerate the model's training process but also enhance the model's overall performance. By training for only 2 epochs, our method achieved improvements of 42.6, 3.4, and 3.3 percentage points in the AUPRO, AP, and $F_1$-max metrics, respectively. These significant enhancements further validate the effectiveness and advantages of using fuzzy prompts in anomaly detection tasks.

Table 5: The impact of different prompts and data settings on pixel-level results.

| Prompt | Data Settings | Pixel-level | | | #Imgs Num |
|---|---|---|---|---|---|
| | | AUPRO | AP | $F_1$-max | |
| Precise | All Data | 44.0 | 40.8 | 43.3 | 2162 |
| Stacked | All Data | **86.6** | 44.2 | 46.6 | 2162 |
| | Cluster1 | 86.1 | 44.3 | 46.6 | 1360 |
| | Cluster2 | 84.1 | 39.9 | 43.4 | 802 |
| | EFA | 86.4 | **46.0** | **47.6** | 2162 |

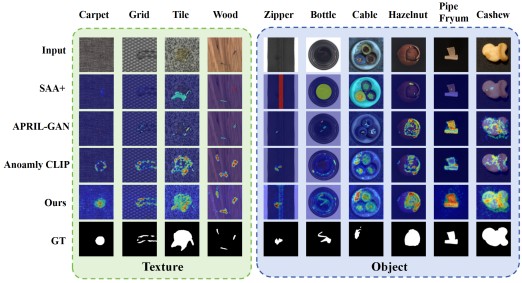

Figure 3: Comparison of visualization results

### 3.3.2 THE EFFECTIVENESS OF CSP AND EFA

In our framework, the Clustering-Driven Stacked Prompt (CSP) and Ensemble Feature Alignment (EFA) modules work together to accomplish the anomaly detection and segmentation tasks. Specifically, in the CSP module, we employ the K-means clustering method to select classes and determine the number of categories through our designed scoring mechanism. The figure Fig 4 below clearly illustrates the classification results on two public datasets: MVTec was categorized into one class, while VisA was divided into two classes.

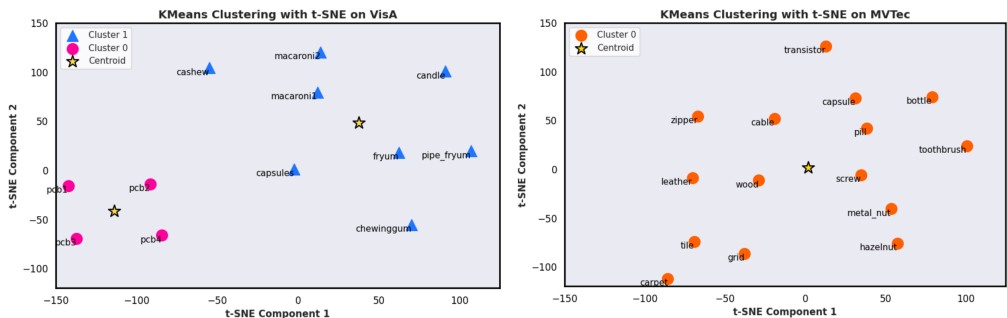

Figure 4: Clustering result of the MVTec and VisA datasets after the Clustering-Driven Stacked Prompt (CSP) module.

In the subsequent experiments, we removed the CSP module, which involved training a single set of linear layers directly using all categories to obtain a baseline result. Then, we introduced the CSP module and conducted training on the VisA dataset. The CSP module divided the dataset into two categories: one category included pcb1, pcb2, pcb3, pcb4, while the other category contained the remaining classes. We trained on these two categories separately and tested each using the corresponding trained linear layers during the testing phase.

Surprisingly, as shown in Table 5, under the Cluster1 and Cluster2 settings, despite a significant reduction in training data, the results were comparable to those of the model trained with the full dataset. Particularly in the Cluster1 setting, the AP metric even slightly surpassed our baseline results. This indicates that, even with reduced training data, the clustering strategy implemented through the CSP module can effectively enhance the model's performance, especially when the data distribution is relatively clear. By incorporating the EFA module, our approach dynamically assigns different weights to the outputs of various linear layers based on the distances between test samples and multiple clustering centers. This adaptive weighting method ensures that the strengths of each linear layer are effectively utilized. As a result, the model becomes more flexible and accurate in handling anomaly detection segmentation tasks, particularly when dealing with diverse categories.

We believe that the limited sample size and number of categories in the MVTec and VisA datasets restrict the performance of our method. Therefore, we merged multiple datasets and categorized them according to our approach for comparison. The results are shown in Table 6. As the number of clusters increases, all metrics improve. For certain metrics that show a decline, we analyzed the

Table 6: Pixel-Level Performance Across Datasets

| Train \ Test | | MVTec | VisA | DTD | MPDD | DAGM | BTAD | SDD | Average |
|---|---|---|---|---|---|---|---|---|---|
| Train Data | Cluster Num | | | | | | | | |
| VisA | 1 | (86.6,44.2,46.6) | (—,—,—) | (90.7,66.0,65.2) | (88.5,28.6,33.5) | (80.2,44.3,48.6) | (79.3,47.7,51.8) | (88.1,14.4,27.6) | (85.6,40.9,45.6) |
| | 2 | (86.4,46.0,47.6) | (—,—,—) | (91.3,66.4,65.1) | (89.1,31.2,35.1) | (79.3,45.9,50.0) | (76.0,48.6,53.5) | (88.0,22.1,35.4) | (85.0,43.4,47.8) |
| MVTec_MPDD | 1 | (—,—,—) | (89.4,25.7,32.0) | (92.5,68.4,66.9) | (—,—,—) | (81.1,48.3,52.1) | (73.5,37.1,42.4) | (93.7,41.9,47.3) | (86.0,44.3,48.1) |
| | 2 | (—,—,—) | (88.9,22.8,30.0) | (91.9,70.3,68.9) | (—,—,—) | (79.4,49.6,53.4) | (76.1,40.8,44.4) | (94.1,44.2,50.1) | (86.1,45.5,49.4) |
| MVTec_DTD | 1 | (—,—,—) | (87.8,23.6,30.4) | (—,—,—) | (85.9,27.4,33.4) | (77.8,49.3,54.6) | (68.7,32.0,37.2) | (89.6,44.1,51.0) | (82.0,35.3,41.3) |
| | 2 | (—,—,—) | (88.4,20.5,27.5) | (—,—,—) | (88.3,26.6,33.3) | (79.7,54.0,57.0) | (72.0,28.0,33.9) | (92.9,39.3,48.4) | (84.3,33.7,40.0) |
| | 3 | (—,—,—) | (89.1,23.9,30.8) | (—,—,—) | (87.6,26.5,33.0) | (80.6,53.3,56.1) | (73.4,32.7,37.5) | (92.9,41.5,50.3) | (84.7,35.6,41.5) |
| MVTec_DAGM | 1 | (—,—,—) | (90.0,22.7,30.1) | (94.2,75.6,73.6) | (89.5,27.2,34.1) | (—,—,—) | (78.9,42.4,47.7) | (93.2,41.0,51.5) | (89.2,41.8,47.4) |
| | 2 | (—,—,—) | (90.1,21.5,30.1) | (94.2,71.9,72.9) | (89.7,26.2,32.4) | (—,—,—) | (81.2,45.7,50.3) | (95.4,42.4,51.6) | (90.1,41.5,47.5) |
| | 3 | (—,—,—) | (90.1,23.1,30.4) | (94.9,76.6,74.0) | (90.4,28.1,34.8) | (—,—,—) | (81.7,46.1,51.6) | (95.3,42.8,50.1) | (90.5,43.3,48.2) |

situation and found that this is primarily due to significant data shifts during K-means clustering, leading to underfitting in the linear layers and thus affecting the overall performance results.

### 3.3.3 THE EFFECTIVENESS OF RPL

In our framework, the Regulating Prompt Learning (RPL) module enhances anomaly detection classification performance through regularization with fuzzy stacked prompts. We first evaluated the effectiveness of using fuzzy stacked prompts for regularization.

Table 7 shows a comparison between using only the classification loss ($\mathcal{L}_{cls}$) and incorporating regularization loss ($\mathcal{L}_{text}$) with fuzzy stacked prompts into the classification loss. We observed a significant improvement in anomaly detection classification performance with the inclusion of fuzzy stacked prompt regularization loss. This indicates that the regularization method plays a crucial role in enhancing the

| Setting | Image-level (VisA) | | |
|---|---|---|---|
| | AUROC | AP | $F_1$-max |
| w/o RPL | 79.0 | 82.9 | 78.8 |
| w RPL | **84.7** | **86.9** | **82.7** |

Table 7: RPL module ablation study

model's generalization capability and improving classification performance. In addition, several key factors significantly impact the performance of prompt learning, including Depth of Learnable Token Embeddings M; Length of Learnable Token Embeddings L; Learnable text prompts E; Initialization of Prompts; Detailed analysis can be seen in appendix A

## 4 CONCLUSION

In this paper, we propose the concept of fuzzy stacked prompts and apply it to industrial anomaly detection. This pre-training method is not only simple and efficient but also significantly enhances the classification and segmentation capabilities in anomaly detection. Additionally, our approach can continually improve anomaly detection performance in real industrial scenarios by adding more linear layers as the data volume and number of categories increase. However, our method also has some limitations. For instance, the performance may not meet expectations for categories that are difficult to describe accurately with text. In future work, we will continue to explore ways to further enhance the feature alignment capabilities of CLIP to address more complex scenarios.

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

## A    APPENDIX

This supplementary appendix includes the following contents: A.1 a more detailed introduction to some state-of-the-art methods; A.2 an analysis of some hyperparameters of the RPL module; A.3 an analysis of how to select categories from text prompts during the segmentation process; A.4 ablation experiments on the selection of template initialization in the prompt learning process; A.5 presentation of details of some results.

### A.1    STATE-OF-THE-ART METHODS.

- **WinCLIP** (Jeong et al., 2023) They create an extensive collection of custom-designed text prompt templates tailored for anomaly detection and employ a window scaling strategy to achieve anomaly segmentation. This method efficiently accomplishes anomaly detection, segmentation, and classification tasks by extracting features at different scales.

- **APRIL-GAN** (Chen & et al., 2023) APRIL-GAN is an enhanced version of WinCLIP. It first optimizes the text prompt templates and then enhances local visual semantics by combining learnable linear projections. Through the design of linear layers, it strengthens the alignment between image features and text features at different levels, thus achieving more precise segmentation.

- **CLIP-AD** (Chen et al., 2023) CLIP-AD utilizes a text prompting design similar to Win-CLIP, adapting to ZSAS tasks through multi-branch feature surgery design and fine-tuning techniques.

- **CoCoOp** (Zhou et al., 2022) CoCoOp is a method that applies CLIP to image classification tasks based on prompt learning. It uses continuously learnable vectors instead of manually designed text prompts, enhancing the model's generalization to novel classes by making the prompt conditioned on each input image. To adapt CoCoOp to the ZSAS task, we improved the prompt templates used in the original paper. Specifically, the original template $[v_1(x)][v_2(x)]\cdots[v_r(x)][\text{class}]$ is replaced with $[v_1(x)][v_2(x)]\cdots[v_r(x)][\text{good}][\text{class}]$ and $[v_1(x)][v_2(x)]\cdots[v_r(x)][\text{damaged}][\text{class}]$ for the generation of normal and abnormal text prompts, where $v_i(x)$ represents the learnable word embeddings that incorporate image features $x$.

- AnomalyCLIP (Zhou et al., 2024) AnomalyCLIP proposes learning object-agnostic textual prompts for zero-shot anomaly detection. It replaces specific product categories [class] with [object] in the textual prompts, enabling the model to focus on the anomalous regions in the images.

## A.2 HYPARAMETER ANALYSIS.

We studied the impact of the depth $M$ of learnable token embeddings, the length $L$ of learnable token embeddings, and the learnable text prompt $E$ on model performance, with evaluation metrics including AP, $F_1$-max, and AUROC, as shown in Fig 5. The trends for all metrics are generally consistent. Since we employed the Regulating Prompt Learning (RPL) module, the learnable token length can reach 20, the depth can reach 9, and the learning length of the text prompts is set to 12. Under these parameters, the trends for all evaluation metrics are consistent: when the parameters have not reached optimal values, the model is in an underfitting state and cannot effectively detect anomalies; when the parameters exceed the optimal values, the model enters an overfitting state, learning some redundant information that negatively impacts the final results. This highlights the importance of carefully selecting parameters during training to ensure that the model effectively captures useful anomaly features.

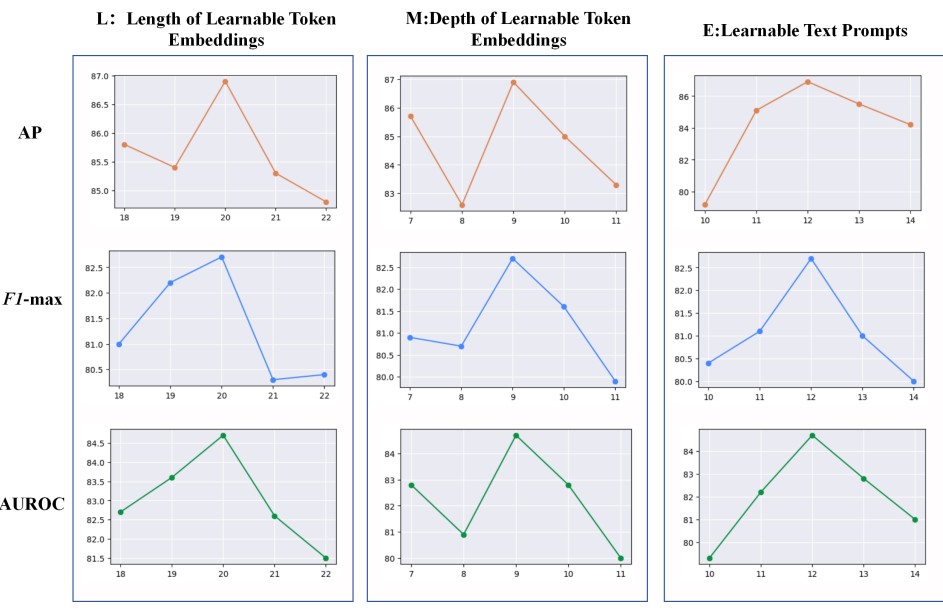

Figure 5: Hyperparameter Analysis. We present the Length of Learnable Token Embeddings L; the Depth of Learnable Token Embeddings M; Learnable text prompts E ablation study of the AP, $F_1$-max, and AUROC across the dimensions of Length of Learnable Text Prompt, Depth of Learnable Text Prompt, and Prompt Length. The orange represents AP, the blue represents $F_1$-max, and the green represents AUROC.

## A.3 ANALYSIS OF TEST CATEGORY NAMES SELECTION.

During the segmentation testing, it is crucial to select appropriate textual prompts. To adapt to our proposed Ensemble Feature Alignment (EFA) module, we conducted ablation experiments on different prompts, with the results shown in Table 8. Initially, we tested using only the test object names and only the stacked object names during training, finding that the difference between these two approaches was minimal. Subsequently, combining the two yielded significant effects. Using test object names enhanced the perceptual capabilities of the CLIP model, making it more focused on the target rather than the background; while the stacked object names used during training better aligned with the trained linear layer, guiding the model to focus on anomalies rather than complete objects. We also tested the combination of using both test object names and stacked object names

during training, but the results were not as effective as the previous method. This indicates that repeatedly mentioning object names may cause CLIP to shift its focus on the target, thereby affecting the model's performance. This experimental result underscores the importance of avoiding redundant information when selecting textual prompts to maintain the model's sensitivity to anomalies.

Table 8: Comparison of the selection of class in abnormal segmentation tests.

| Test prompt | AUROC | AP | $F_1$-max |
|---|---|---|---|
| Test_obj | **89.8** | 26.5 | 33.0 |
| cluster | **89.8** | 26.5 | 32.9 |
| Test_obj+cluster | **89.8** | **28.0** | **34.2** |
| Test_obj+cluster+Test_obj | 89.7 | 26.9 | 33.0 |

A.4 PROMPT LEARNING INITIALIZATION TEMPLATE SELECTION ANALYSIS.

In prompt learning, template initialization is a crucial influencing factor. We conducted tests on various templates while maintaining the design of stacked fuzzy prompts, where "cluster" represents stacked fuzzy prompts. To simplify the experiment, we default the number of clusters to 1. In addition to stacked prompts, we also tested abstract prompts, using "object" instead of specific category names. The results showed that the overall performance was not as good as the initialized stacked fuzzy prompts. This indicates that stacked fuzzy prompts can more effectively capture exceptional features during initialization, enhancing the performance of model.

Table 9: The experimental results when using different text prompt templates during prompt learning

| Test on VisA | AUROC | AP | $F_1$-max |
|---|---|---|---|
| a photo of a good [**cluster**]
a photo of a damaged [**cluster**] | 82.6 | 85.9 | 81.0 |
| This is a good photo of [**cluster**]
This is a damaged photo of [**cluster**] | 82.5 | 85.9 | **81.1** |
| It is a photo of a [**cluster**] without damage
It is a photo of a [**cluster**] with damage | 79.7 | 83.0 | 80.4 |
| There is not a damaged [**cluster**] in the photo
There is a damaged [**cluster**] in the photo | **82.9** | **86.1** | **81.1** |
| It is a good, perfect and pristine picture of [**cluster**]
It is a damaged, flawed, and broken picture of [**cluster**] | 81.7 | 85.1 | 80.3 |
| a photo of a good **object**
a photo of a damaged **object** | 81.2 | 83.9 | 80.8 |
| This is a good photo of **object**
This is a damaged photo of **object** | 80.3 | 83.4 | 80.6 |
| It is a photo of a **object** without damage
It is a photo of a **object** with damage | 78.0 | 81.4 | 79.7 |
| There is not a damaged **object** in the photo
There is a damaged **object** in the photo | 81.8 | 85.1 | **81.1** |
| It is a good, perfect and pristine picture of **object**
It is a damaged, flawed, and broken picture of **object** | 79.1 | 82.8 | 79.8 |

A.5 FINE-GRAINED ZSAD PERFORMANCE.

In this section, we present the fine-grained data subset-level ZSAD performance in detail.

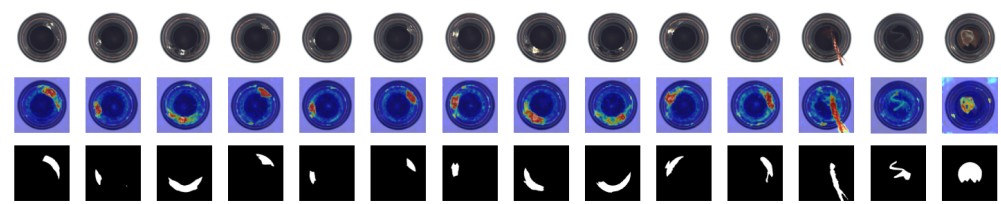

Figure 6: Anomaly score maps for the data Bottle. The first row represents the input. The second row presents the segmentation results from FuzzyCLIP. The last line is the ground truth.

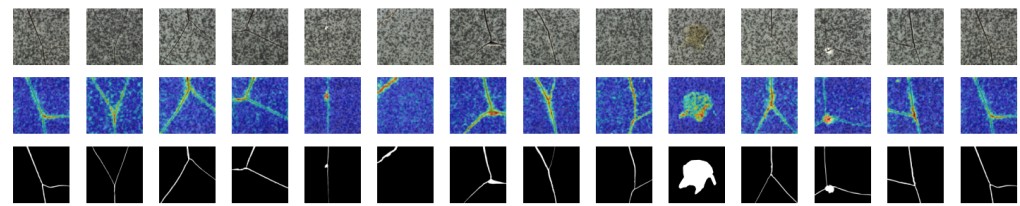

Figure 7: Anomaly score maps for the data Tile. The first row represents the input. The second row presents the segmentation results from FuzzyCLIP. The last line is the ground truth.

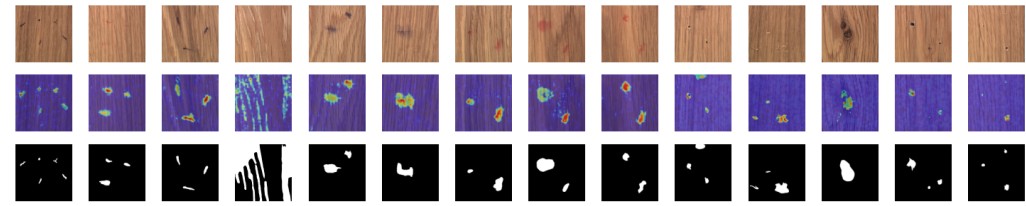

Figure 8: Anomaly score maps for the data Wood. The first row represents the input. The second row presents the segmentation results from FuzzyCLIP. The last line is the ground truth.

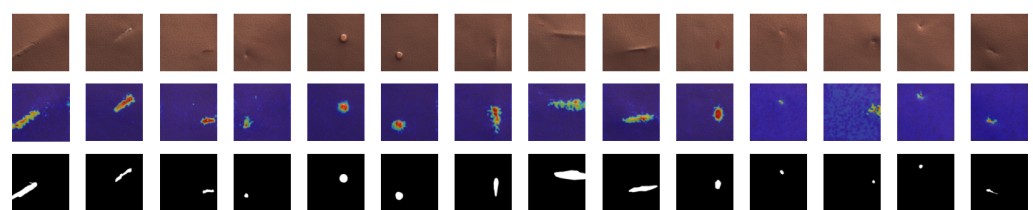

Figure 9: Anomaly score maps for the data Leather. The first row represents the input. The second row presents the segmentation results from FuzzyCLIP. The last line is the ground truth.

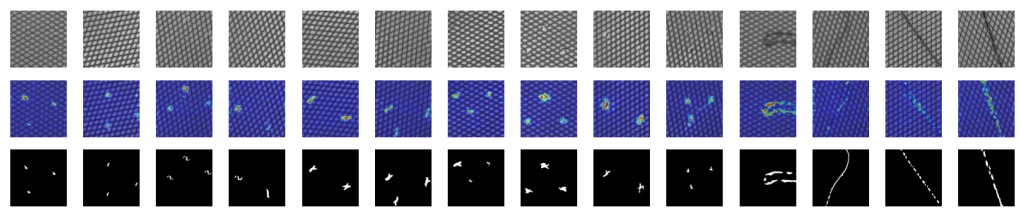

Figure 10: Anomaly score maps for the data Grid. The first row represents the input. The second row presents the segmentation results from FuzzyCLIP. The last line is the ground truth.

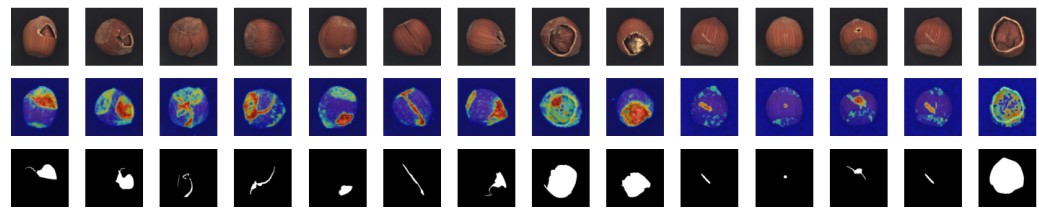

Figure 11: Anomaly score maps for the data Hazelnut. The first row represents the input. The second row presents the segmentation results from FuzzyCLIP. The last line is the ground truth.

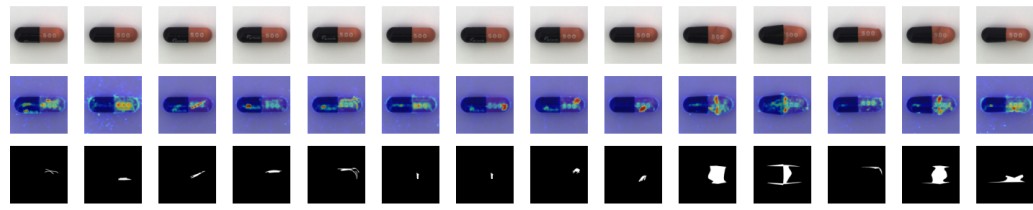

Figure 12: Anomaly score maps for the data Capsule. The first row represents the input. The second row presents the segmentation results from FuzzyCLIP. The last line is the ground truth.

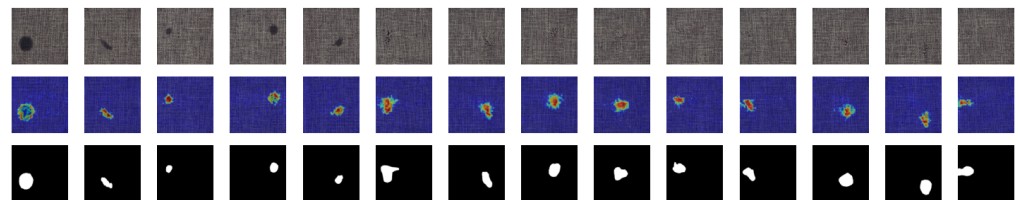

Figure 13: Anomaly score maps for the data Carpet. The first row represents the input. The second row presents the segmentation results from FuzzyCLIP. The last line is the ground truth.

