# OpenReview forum: "FuzzyCLIP: Clustering-Driven Stacked Prompt in Zero-Shot Anomaly Detection"
_ICLR.cc/2025/Conference — ICLR 2025 Conference Withdrawn Submission_

### Official Review · Reviewer_oYQM · 2024-10-21

**Soundness:** 3
**Presentation:** 2
**Contribution:** 2
**Rating:** 5
**Confidence:** 5

**Summary:**

This paper analyzes the problems of linear adapter-based zero-shot anomaly detection methods, and points out that the traditional prompt design has strong training fluctuations and overfitting problems. To solve the above problems, the authors proposed fuzzy prompts and Ensemble Feature Alignment. In addition, this paper introduced regulating prompt learning, using fuzzy stacked prompts to constrain prompt learning to complete anomaly classification.

**Strengths:**

1. The proposed fuzzy prompt design makes the pre-training process of zero-shot anomaly detection more stable and improves the performance.
2. The experimental results show that this work has performance improvement on some data sets.

**Weaknesses:**

1. The writing logic of the paper is confusing. In Figure 1, c does not reflect clustering and EFA, and line 094 mentions that "After clustering training, different linear layers..." making the article very disorganized. Moreover, the prompt design of the test phase in Figure 1c and Figure 2 do not correspond.
2. The experimental results show that the overall performance of the method does not seem to be good on all datasets
3. See the question for more details.

**Questions:**

1. The using of notation is in confusion, “Clsa... Clsn” in equation (3) “clsA...clsB” and “clsA1... clsAn, clsB1... clsBn” in Figure 2, what do they mean and how do they relate to each other the author needs to explain.
2. Is the clustering prompt in the test phase the same as the clustering prompts in the training phase?
3. Does the variable x in equation (4) refer to the features of the prompt in equation (1)?
4. I would like to know the experimental results of Anomaly Classification with Clustering-Driven Stacked Prompt instead of learnable Prompt in RPL.
5. The ablation experiments for CSP and EFA need to be supplemented.
6. What are the experimental results of replacing class names with random characters in the clustering process?

---

### Official Review · Reviewer_1Mhq · 2024-11-02

**Soundness:** 3
**Presentation:** 3
**Contribution:** 3
**Rating:** 8
**Confidence:** 3

**Summary:**

This work discusses enhancing text and image alignment in the CLIP model for zero-shot anomaly detection. It introduces fuzzy prompts, Clustering-Driven Stacked Prompts (CSP), and Ensemble Feature Alignment (EFA) to improve alignment and generalization. The approach outperforms others in training speed, stability, and convergence, particularly in anomaly detection segmentation. Fuzzy stacked prompts show strong generalization in classification tasks without extra operations. The Regulating Prompt Learning (RPL) module further leverages this generalization for improved anomaly detection classification. Extensive experiments on seven industrial anomaly detection datasets demonstrate state-of-the-art performance in zero-shot anomaly detection and segmentation.

**Strengths:**

1. The point view of the fuzzy prompts provides a concise idea to improve the model's performance and provide a general method for different objects.
2. CFP, EFA, and RPL provide a comprehensive method for both the segmentation and classification tasks, which is a more precise design for the ZASA.
3. The experiments are comprehensive and show the effectiveness of their method.

**Weaknesses:**

1. The clustering method is confused. In Fig 4, VisA contains 2 clusters and MVTec only contains one cluster, will this small number of clusters improve the technique rather than just using the average of these features?
2. In Fig 3, compared to April-GAN and AnomalyCLIP, it seems that your methods will generate some noise on the segmentation result, what masks this noise, and will it influence the result in real-world applications?

**Questions:**

See Weakness.

---

### Official Review · Reviewer_9tDc · 2024-11-03

**Soundness:** 1
**Presentation:** 1
**Contribution:** 2
**Rating:** 3
**Confidence:** 4

**Summary:**

This study proposes the FuzzyCLIP, applying Clustering-Driven Stacked Prompts (CSP) with Ensemble Feature Alignment (EFA) and Regulating Prompt Learning (RPL) to improve image-text alignment in CLIP for Zero-shot Anomaly Detection (ZSAD).

**Strengths:**

A novel cluster-based stacked prompt approach is proposed, setting it apart from conventional prompt methods.

**Weaknesses:**

1. The paper includes an introduction section, but it lacks the title "Introduction."
2. Several aspects of Figures 1 and 2 are unclear:
  - Figure 1: Please specify which method's training process generated the performance graphs in Figure 1 (a) and (b). Adding explicit labels or annotations indicating the methods shown, along with details on the experimental setup, would clarify the results presented.
  - Figure 2: To improve clarity in Figure 2, consider adding labels or arrows to distinguish the inputs, operations, and outputs within the EFA module. Currently, the output seems to be multiplied by the features, yet this process is also labeled as EFA, which may be confusing.
3. Additionally, the equations require clarification:
  - In Equation 4, the method for finding the optimal class number K* seems to select a specific class K with the smallest distance difference, which could be misinterpreted as choosing a particular class k rather than determining the number of clusters k. Additionally, definitions for n, x, and x' are missing.
4. Finally, there are some limitations regarding the effectiveness demonstrated in the experiments.
  - For Tables 2 and 3, consider either using a consistent set of comparison methods across all datasets or explaining why different methods were chosen for each dataset. This would help readers interpret the results more accurately.

Further specific questions are provided in the Questions section.

**Questions:**

1. Figure 2: RPL is trained to approximate stacked prompts. I am curious about the distinction between directly applying fuzzy prompts and using RPL for this purpose.
2. Figure 4: Although the proposed methodology is described as clustering-driven, I noticed that only one cluster is formed in MVTecAD. This observation raises the question of whether clustering is meaningful in this context.
3. Extending from Question 2: Table 5 suggests that the entire dataset can be represented by learning either the full dataset or specific clusters within an optimal number of 𝐾 clusters. The authors argue that the expressiveness of individual clusters is demonstrated by comparing these two approaches. However, this claim seems limited without experiments on datasets with a large number of classes and clusters. It is unclear whether the high ZSAD performance results from the explanatory power of each cluster’s data or merely from the strength of the pre-trained backbone, which may require minimal data for effective learning.

---

### Official Review · Reviewer_RfEM · 2024-11-05

**Soundness:** 3
**Presentation:** 3
**Contribution:** 3
**Rating:** 5
**Confidence:** 5

**Summary:**

The paper introduces a novel approach for zero-shot anomaly detection in industrial settings, leveraging the CLIP model. The authors propose a concept of fuzzy prompts and introduce Clustering-Driven Stacked Prompts (CSP) along with an Ensemble Feature Alignment (EFA) module to improve the alignment between text and image features. They also introduce a Regulating Prompt Learning (RPL) module to regularize prompt learning, enhancing classification performance. The authors claim that their method achieves state-of-the-art performance in zero-shot anomaly detection and segmentation tasks across seven industrial anomaly detection datasets.

**Strengths:**

1. Interesting Concept: The paper introduces an interesting concept by focusing on internal clustering characteristics, which could provide useful insights for future research in anomaly detection.

2. Clear Presentation: The paper is well-structured, making it easy to follow the methodology and results.

3. Relevant Experiments: The choice of datasets reflects a good range of real-world scenarios.

**Weaknesses:**

1.Insufficient Innovation: While the paper introduces the concept of fuzzy prompts and a clustering-driven approach, the fundamental techniques used are extensions of existing methods in the field of vision-language models and prompt learning. The novelty might be seen as limited because it primarily combines known strategies in a new way rather than presenting a groundbreaking new algorithm or theoretical framework. This could lead to questions about the true advancement the paper offers to the field, beyond the specific application to zero-shot anomaly detection.

2.Dependence on Dataset Characteristics: The effectiveness of the method seems to be highly dependent on the dataset's characteristics, particularly the clarity of internal clustering boundaries. For instance, the method performs better on the VisA dataset, where the clustering boundaries are clear, compared to the MVTec dataset, where it struggles to distinguish multiple clusters.

3.Insufficient Experimental Evidence: The experimental results may not be sufficient to fully support all the claims made in the paper. Specifically, in two out of the five datasets, their method does not outperform AnomalyCLIP, which raises questions about the general applicability of their approach.

Minor issues:
Additionally, the paper exhibits a lack of consistency in the referencing of figures and tables. While figures are abbreviated as "Fig", tables lack any abbreviation. Furthermore, the abbreviation for figures should be standardized to "Fig." with a period, to adhere to conventional academic writing standards. Besides, futher descriptions for tables and figures should be added to the captions respectively.

**Questions:**

Please kindly refer to the weakness.

---

### Note · Authors · 2024-11-15

I have read and agree with the venue's withdrawal policy on behalf of myself and my co-authors.